# Delay-dependent contributions of medial temporal lobe regions to episodic memory retrieval

**Maureen Ritchey[1]\*, Maria E Montchal[1], Andrew P Yonelinas[2], Charan Ranganath[1,2]**

[1]Center for Neuroscience, University of California, Davis, Davis, United States; [2]Department of Psychology, University of California, Davis, Davis, United States

**Abstract** The medial temporal lobes play an important role in episodic memory, but over time, hippocampal contributions to retrieval may be diminished. However, it is unclear whether such changes are related to the ability to retrieve contextual information, and whether they are common across all medial temporal regions. Here, we used functional neuroimaging to compare neural responses during immediate and delayed recognition. Results showed that recollection-related activity in the posterior hippocampus declined after a 1-day delay. In contrast, activity was relatively stable in the anterior hippocampus and in neocortical areas. Multi-voxel pattern similarity analyses also revealed that anterior hippocampal patterns contained information about context during item recognition, and after a delay, context coding in this region was related to successful retention of context information. Together, these findings suggest that the anterior and posterior hippocampus have different contributions to memory over time and that neurobiological models of memory must account for these differences.

## Introduction

The medial temporal lobes (MTL) are known to play a key role in the formation of lasting memories, but there has been considerable debate about whether their involvement in memory retrieval is stable over time. Some models have suggested that the hippocampal formation (HF) is critical for supporting new memories, but that, over time, memories can be supported by neocortical areas alone. In one account, the shift from hippocampal to cortical representation reflects a transfer of the memory trace through a time-dependent process known as systems consolidation (*Squire, 1992*; *Alvarez and Squire, 1994*). This type of transfer is thought to preserve the quality and contents of the memory, although older memories may generally be weaker. Other accounts, however, have argued that changes in hippocampal vs cortical involvement are accompanied by the transformation of episodic memories, which require the HF, into semantic memories, which lack episodic context information and can be supported by cortex alone (*Nadel and Moscovitch, 1997*; *Winocur et al., 2010*).

Changes in the neural bases of memory have typically been described as unfolding over very long timescales, but some studies have documented changes even across relatively short delays. For instance, some functional magnetic resonance imaging (fMRI) studies have shown that, when both encoding and retrieval are controlled in the laboratory, retrieval-related activity in the HF declines from immediate to delayed test with even a 1-day delay (*Bosshardt et al., 2005a*; *Takashima et al., 2006, 2009*; but see *Stark and Squire, 2000*). These findings are echoed by work demonstrating that a night of sleep, or even a brief nap, can alter the neural bases of memory (*Takashima et al., 2006*; *Gais et al., 2007*; *Sterpenich et al., 2007*) and have long-term consequences for memory (*Diekelmann and Born, 2010*). Although these findings are often interpreted as reflecting the early stages of memory systems consolidation, it has remained a challenge to separate changes in neural representation

\*For correspondence: meritchey@ucdavis.edu

**Competing interests:** The authors declare that no competing interests exist.

**eLife digest** In 1953, an American man called Henry Molaison underwent surgery to remove the medial temporal lobes of his brain in an effort to cure him of severe epilepsy. After the surgery, his epilepsy was indeed improved, but he was left without the ability to form new memories. His case is now seen as one of the first demonstrations of the medial temporal lobes being involved in memory.

Beneath the surface of each medial temporal lobe is a structure called the hippocampus, which is essential for the formation of new memories. However, memories are not stored permanently within the hippocampus: instead they are transferred ultimately to the neocortex, which is the outer layer of the brain.

Some neuroscientists believe that the content of memories remains unchanged during this transfer, whereas others argue that memories are stripped of their context—that is, details of when and where they were acquired—before they reach the neocortex.

In a brain imaging experiment, Ritchey et al. have now attempted to distinguish between these two possibilities. Volunteers were asked to memorize sentences linking an object to a room, such as 'the apple is in the bedroom', on two occasions 24 hr apart. Immediately after the second session, the volunteers were asked to complete memory tests while lying in the brain scanner. In one test the volunteer was shown a list of objects and asked to identify those objects they could recall seeing in either of the training sessions, and to identify objects they recognised as familiar, even if they could not specifically remember seeing these objects during training sessions.

Analysis of the brain imaging data revealed that regions of the medial temporal lobes were more active when the volunteers recalled objects than when they recognised them as familiar. Moreover, for the 'recall' responses—in which the volunteers could still retrieve contextual information—the activity of the hippocampus depended on the age of the memories. The anterior (front) part of the hippocampus was active when subjects recalled either new memories or memories from 24 hr previously, whereas the posterior (rear) hippocampus was active only during the recall of new memories. In addition, patterns of activity observed in the anterior hippocampus could be used to determine which room was previously associated with the object. This suggests that contextual information is retained in the anterior hippocampus, but lost from the posterior hippocampus over time.

Overall the results of Ritchey et al. highlight the fact that individual components of the medial temporal lobes, including hippocampal subregions, have distinct roles in the storage of memories, with these roles also changing over time. Moreover, the data lend support to the idea that contextual information may be lost from memories before they reach the neocortex.

from concomitant changes in episodic quality or content. For instance, one study found that differences in HF activity for recent vs remote autobiographical retrieval could be explained by differences in memory vividness (*Gilboa et al., 2004*). One way to control for these differences is to limit analysis to memories endorsed with high confidence or recollection (*Takashima et al., 2006*; *Sterpenich et al., 2009*; *Takashima et al., 2009*; *Milton et al., 2011*). However, even this approach could be insensitive to differences in the kinds of details that accompany recollection, such as information related to episodic context. The HF is especially involved in tasks that require retrieval of contextual details (*Davachi, 2006*; *Eichenbaum et al., 2007*; *Montaldi and Mayes, 2010*; *Ranganath, 2010*), so differences between HF and cortical contributions to memory over time could be due to changes in contextual retrieval.

Another challenge to understanding changes in the neural bases of memory is that current models have not accounted for heterogeneity of function within the MTL. In particular, the perirhinal cortex (PRC) and parahippocampal cortex (PHC) are critically involved in episodic memory, yet they affiliate with different large-scale cortical networks (*Libby et al., 2012*; *Ranganath and Ritchey, 2012*; *Ritchey et al., 2014*) and are widely believed to be functionally distinct from each other and from the HF (*Davachi, 2006*; *Eichenbaum et al., 2007*; *Montaldi and Mayes, 2010*; *Norman, 2010*; *Ranganath, 2010*). Nonetheless, current models have been vague with respect to their predictions for the PRC and

PHC, either grouping them alongside the HF (*Squire, 1992*; *McClelland et al., 1995*; *Nadel and Moscovitch, 1997*) or ignoring them altogether (*Winocur et al., 2010*). The PRC and PHC also functionally interact with different pathways along the longitudinal axis of the HF (*Kahn et al., 2008*; *Poppenk and Moscovitch, 2011*; *Libby et al., 2012*), suggesting additional heterogeneity within the HF. For instance, the anterior and posterior HF are thought to play different roles in memory for spatial context (*Moser and Moser, 1998*; *Fanselow and Dong, 2010*; *Poppenk et al., 2013*), with coarse context coding in the anterior HF and specific place coding in the posterior HF (*Poppenk et al., 2013*; *Evensmoen et al., 2014*). Because memories can lose contextual specificity over time (*Wiltgen and Silva, 2007*; *Winocur et al., 2007*), differences in the scale of context processing in the MTL might be associated with differences in MTL contributions to memory over time. Despite this heterogeneity, prior imaging studies have not systematically investigated time-dependent differences in recruitment of the anterior and posterior HF, PRC, and PHC during retrieval. Thus, an important next step is to clarify the roles of the anterior and posterior HF and parahippocampal areas in supporting memory over time.

The goal of the present study was to use fMRI to examine changes in MTL activity during immediate and delayed item recollection. Across 2 days, participants encoded sentences, each of which described an association between an item and a room in a house, such that each item was associated with one of eight contexts (*Figure 1A*). Immediately after the second encoding session, participants were scanned while completing an item recognition test. To evaluate responses related to item recollection and context memory, we first compared the overall magnitude of recollection-related activity for each delay, which allowed us to determine whether changes in MTL involvement are observed even when controlling for subjective recollection. Next, we leveraged a novel multi-voxel pattern similarity analysis approach (*Kriegeskorte, 2008*) that measured the sensitivity of voxel patterns in MTL subregions to information about shared study context (c.f., [*Hannula et al., 2013*; *Hsieh et al., 2014*]), thereby providing an objective measure of context reactivation. This analysis allowed us to determine whether a region's continued involvement in recollection is related to its representation of context information in memory. Moreover, we used a region-of-interest (ROI) approach to separately examine the properties of the anterior and posterior HF, PRC, and PHC (*Figure 1B*), thus shedding new light on the regional specificity of memory changes within the MTL.

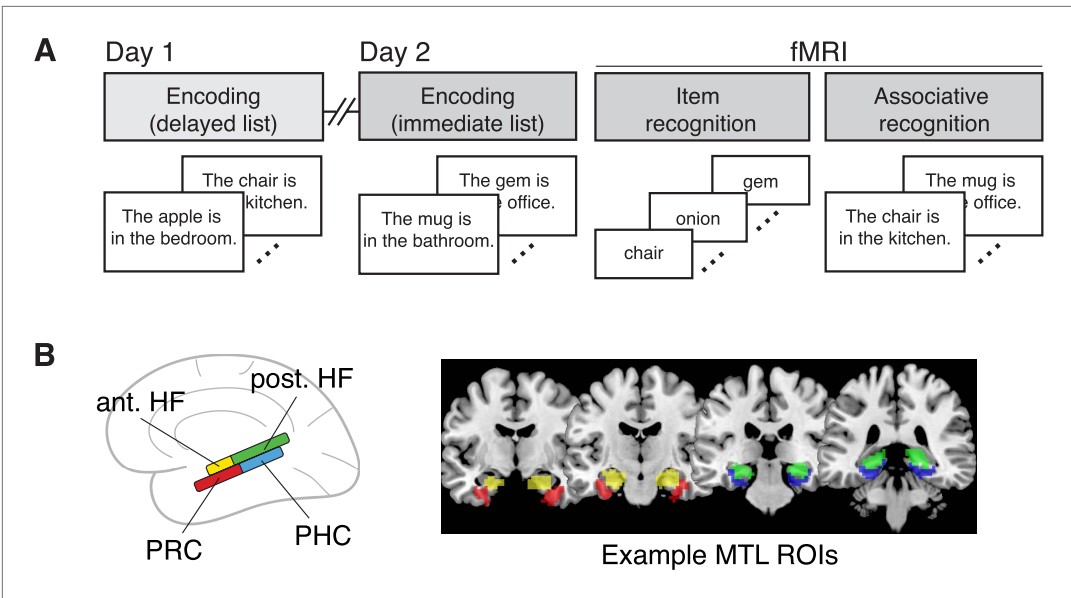

**Figure 1**. Task design and regions of interest. (**A**) Overview of the experimental design. Here, all fMRI analyses are conducted on data from the item recognition phase. (**B**) ROIs included in the main analyses, including the anterior hippocampus (ant. HF), posterior hippocampus (post. HF), perirhinal cortex (PRC), and parahippocampal cortex (PHC). Coronal MRI slices show manually-traced ROIs from a representative subject, resliced to functional resolution and warped into MNI space for display on a template brain.

# Results

## Behavioral results

Behavioral data are presented in *Table 1*. Item memory was evaluated by comparing discriminability for old vs new items in the item recognition test. Item memory was above chance for both immediate, $t(27) = 22.16$, p < 0.001, and delayed test, $t(27) = 18.85$, p < 0.001, and, not surprisingly, accuracy was higher for items tested immediately than after a delay, $t(27) = 8.54$, p < 0.001. This difference was observed for both recollection, $t(27) = 9.51$, p < 0.001, and familiarity, $t(27) = 5.06$, p < 0.001, contributions to item recognition. Context memory was evaluated by comparing discriminability of intact and recombined sentences in the associative recognition test. Participants successfully discriminated between intact and recombined sentences for both immediate, $t(26) = 8.22$, p < 0.001, and delayed test, $t(26) = 8.17$, p < 0.001, and again, associative recognition accuracy was higher for sentences tested immediately than after a delay, $t(26) = 5.13$, p < 0.001. Consistent with the idea that item recollection might involve the reactivation of associative information, associative recognition accuracy was higher for recognized items that were associated with a 'remember' response than for recognized items that were not, $F(1, 26) = 14.20$, p < 0.001. The difference in source accuracy did not interact with delay, $F(1, 26) = 0.27$, p = 0.61.

During the item recognition test, participants were faster to correctly recognize items from the immediate list than from the delayed list, $F(1, 26) = 13.87$, p = 001. Importantly, this difference did not interact with memory status, $F(1, 26) = 0.06$, p = 0.80: that is, the delay effect was observed both for recognized items accompanied by a 'remember' response (immediate: 1.35 ± 0.23 s; delayed: 1.40 ± 0.24 s) and for those that were not (immediate: 1.64 ± 0.35 s; delayed: 1.70 ± 0.29 s). The lack of interaction suggests that response time differences cannot account for delay-dependent activation changes that are specific to recollection. During the associative recognition test, participants were also faster to correctly recognize intact sentences from the immediate list (2.01 + 0.37 s) than from the delayed list (2.24 + 0.51 s), $t(26) = 3.59$, p = 0.001.

## Recollection-related activity during immediate and delayed recognition

We first tested for delay-dependent differences in recollection-related activity during item recognition. Mean activity estimates were extracted from anatomical ROI masks of the anterior HF, posterior HF, PRC, and PHC (*Figure 1B*), then compared with a memory (recollection, familiarity) x delay (immediate, delayed) repeated-measures ANOVA. Within each of these ROIs, activity was greater for recollection than familiarity trials (all $F$s > 4.6, all $p$s < 0.046; *Figure 2*). This recollection effect was stable over time in bilateral PRC, PHC, and anterior HF (all interaction $F$s < 1.67, $p$s > 0.21). However, in the posterior HF, the recollection effect interacted with delay (left: $F(1, 18) = 5.68$, p = 0.028; right: $F(1, 18) = 3.93$, p = 0.063), such that the posterior HF was more active for immediate than delayed

**Table 1.** Behavioral results

| Item recognition | 'R' rate | '4' or '5' rate | d′ | Recollection | Familiarity |
|---|---|---|---|---|---|
| Immediate | 0.46 ± 0.22 | 0.40 ± 0.22 | 1.79 ± 0.43 | 0.44 ± 0.22 | 0.46 ± 0.18 |
| Delayed | 0.25 ± 0.19 | 0.43 ± 17 | 1.14 ± 0.32 | 0.22 ± 18 | 0.32 ± 0.13 |
| Novel | 0.04 ± 0.05 | 0.23 ± 0.09 | – | – | – |

| Associative recognition | 'intact' rate | d′ | % correct for 'R' responses | % correct for '4' or '5' responses |
|---|---|---|---|---|
| Immediate intact | 0.78 ± 0.13 | 1.34 ± 0.59 | 73.5 ± 16.6 | 68.6 ± 14.8 |
| Delayed intact | 0.64 ± 0.17 | 0.59 ± 0.38 | 65.1 ± 18.4 | 57.7 ± 11.4 |
| Immediate recombined | 0.34 ± 0.14 | – | – | – |
| Delayed recombined | 0.43 ± 0.13 | – | – | – |

Note: Summary statistics for individual subjects are contained in *Table 1—source data 1*.

**Source data 1**. Behavioral data.

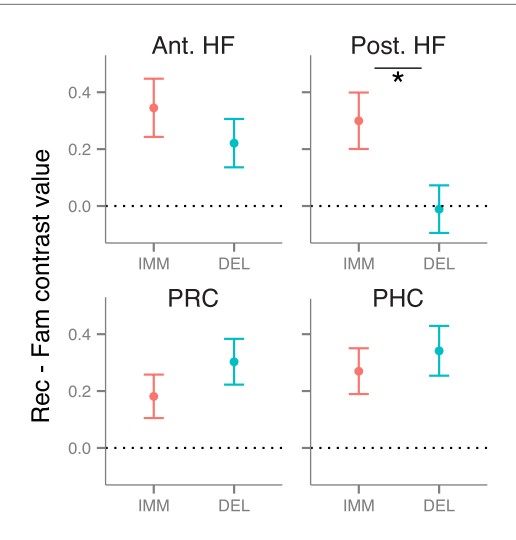

**Figure 2**. Recollection-related activity in the MTL. Univariate estimates of recollection-related activity (i.e., the difference in activation for recollection and familiarity trials) for left hemisphere MTL ROIs (similar results hold for right hemisphere ROIs; see *Figure 2—figure supplement 1*). Asterisk (*) denotes a significant interaction between delay (immediate, delayed) and memory status (recollection, familiarity), p < 0.05. Error bars denote the standard error of the mean. See *Figure 2—figure supplement 2* for results from a model in which the number of recollection and familiarity trials were matched between delays. Summary statistics for individual subjects are contained in *Figure 2—source data 1*, and group-averaged activity estimates for all conditions can be found in *Supplementary file 1*.

The following source data and figure supplements are available for figure 2:

**Source data 1**. Activation estimates for MTL ROIs.

**Figure supplement 1**. Recollection-related activity in right-hemisphere MTL ROIs.

**Figure supplement 2**. Recollection-related activity in a model controlling for the number of trials between delays.

recollection trials (left: $F(18) = 5.61$, p = 0.029; right: $F(18) = 5.75$, p = 0.028), with no concomitant change in familiarity trials, $Fs < 1.26$, $ps > 0.27$. No region showed delay-dependent changes in familiarity estimates (*Supplementary file 1*). The apparent difference in delay effects between the anterior and posterior HF was borne out as an ROI by delay interaction, $F(1, 18) = 5.80$, p = 0.027, indicating that these areas are dissociable on the basis of their contributions in recollection over time.

Because recognition accuracy was higher for items tested immediately than after a delay, findings of delay-dependent differences might be confounded by differences in the numbers of recollection and familiarity trials contributing to the analysis. To control for this potential confound, all comparisons were re-analyzed using a model in which trials were randomly sampled to match numbers of recollection and familiarity across delays. These analyses replicated the MTL ROI findings described above (*Figure 2—figure supplement 2*).

Many studies have shown that recollection is also associated with enhanced activation within an extended neocortical network outside of the MTL (*Spaniol et al., 2009*; *Ranganath and Ritchey, 2012*; *Rugg and Vilberg, 2013*), sometimes referred to as the 'core recollection network' (*Johnson and Rugg, 2007*). To test whether recollection-related activity in this network was modulated by delay, we conducted ROI analyses for the retrosplenial cortex, posterior cingulate, precuneus, angular gyrus, and medial prefrontal cortex. In the left hemisphere, all of these regions showed a main effect of memory, all $Fs > 9.5$, $ps < 0.007$, with no significant interactions with delay, $Fs < 2.21$, $ps > 0.15$ (*Figure 3A*). In the right hemisphere, recollection-related activity in the right precuneus and posterior cingulate declined across the delay (*Figure 3—figure supplement 1*). However, recollection effects in these regions were weaker in general, consistent with previous findings that cortical activity associated with recollection of verbal materials tends to be strongest in the left hemisphere (*Yonelinas et al., 2005*). Exploratory whole-brain, voxel-wise comparisons revealed that both immediate and delayed recollection were associated with activity in the recollection network (corrected p < 0.05; *Figure 3B*), with no significant delay-dependent differences in recollection-related activity (corrected p < 0.05). To better define where recollection-related activity was insensitive to delay, we identified regions that were conjointly involved in immediate and delayed recollection (corrected joint p < 0.05) while excluding voxels showing even small delay differences (liberally defined at p < 0.05 uncorrected). All neocortical regions within the recollection network contained clusters that survived this approach (*Figure 3—figure supplement 2*). These results suggest that, for the most part, recollection-related responses in the neocortical recollection network were maintained across the delay.

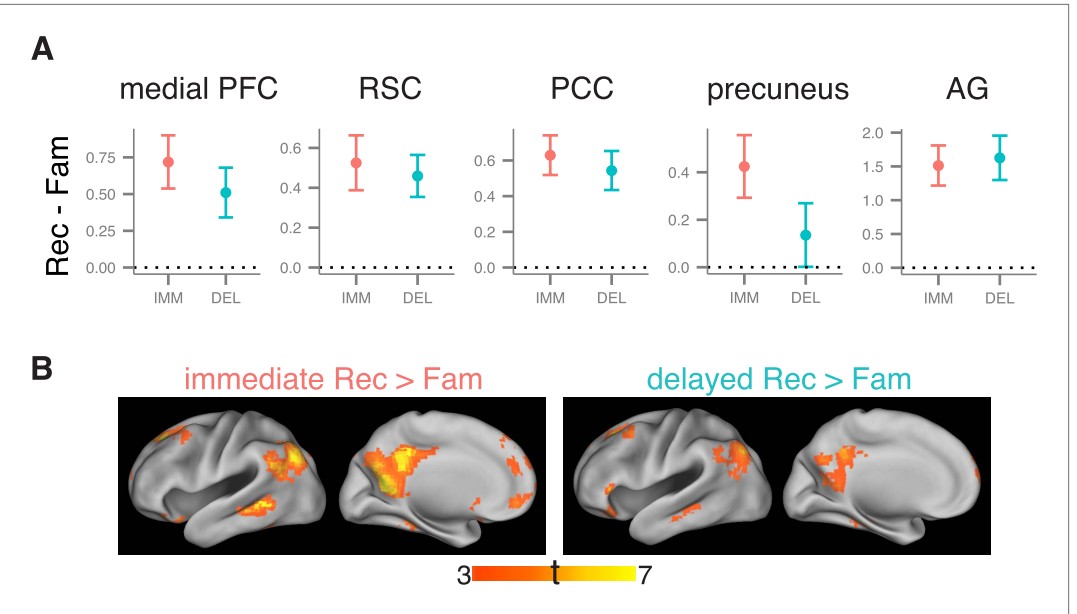

**Figure 3**. Recollection-related activity in the cortical recollection network. (**A**) Univariate estimates of recollection-related activity, that is, the difference in activation for recollection and familiarity trials, for cortical ROIs in the recollection network. Results for left-hemisphere ROIs are shown (see *Figure 3—figure supplement 1* for right-hemisphere ROIs). Error bars denote the standard error of the mean. Note that although the precuneus appears to show a reduction in recollection-related activity over time, the interaction was not significant. Summary statistics for individual subjects are contained in *Figure 3—source data 1*. (**B**) Voxel-wise maps of recollection-related activity, that is, the difference between recollection and familiarity trial activity, thresholded to display significant clusters (voxel-wise p < 0.001, cluster-corrected p < 0.05). Maps are displayed separately for immediate and delayed recollection. Surface images were rendered in Caret using the PALS atlas (left hemisphere shown; see *Figure 3—figure supplement 1* for right hemisphere). Peaks are reported in *Supplementary file 2*. The conjunction of immediate and delayed recollection-related activity is shown in *Figure 3—figure supplement 2*.

The following source data and figure supplements are available for figure 3:

**Source data 1**. Activation estimates for cortical recollection network ROIs.

**Figure supplement 1**. Recollection-related activity in the right hemisphere of the cortical recollection network.

**Figure supplement 2**. Conjunction of immediate and delayed recollection-related activity.

Results from these univariate analyses indicate that differences in brain activity associated with immediate and delayed recollection varied across MTL subregions. The anterior HF and cortical MTL areas maintained their contributions to recollection over time, whereas posterior HF effects were sensitive to delay. Thus, even when memories at both delays were endorsed with recollection, there were changes in posterior HF involvement in memory recognition.

## Context similarity during immediate and delayed recognition

Our next analyses tested whether MTL activity patterns during item recognition carried information about the context (i.e., room) that had been associated with the item at encoding. As depicted in *Figure 4A*, multi-voxel patterns within each ROI were estimated for every recollection trial, and pairs of trials were compared by calculating the similarity between their associated voxel patterns. Similarity values were then summarized according to whether the items had shared context information during encoding (same room from the same study list: for example, 'the apple is in the *bedroom*' and 'the pencil is in the *bedroom*') or had not shared information (different rooms from the same study list: for example, 'the apple is in the *bedroom*' and 'the chair is in the *kitchen*'). Because there was no context information present during the item recognition phase, any pattern similarity differences between these pair types must be ascribed to the reactivation of context information from memory. Thus,

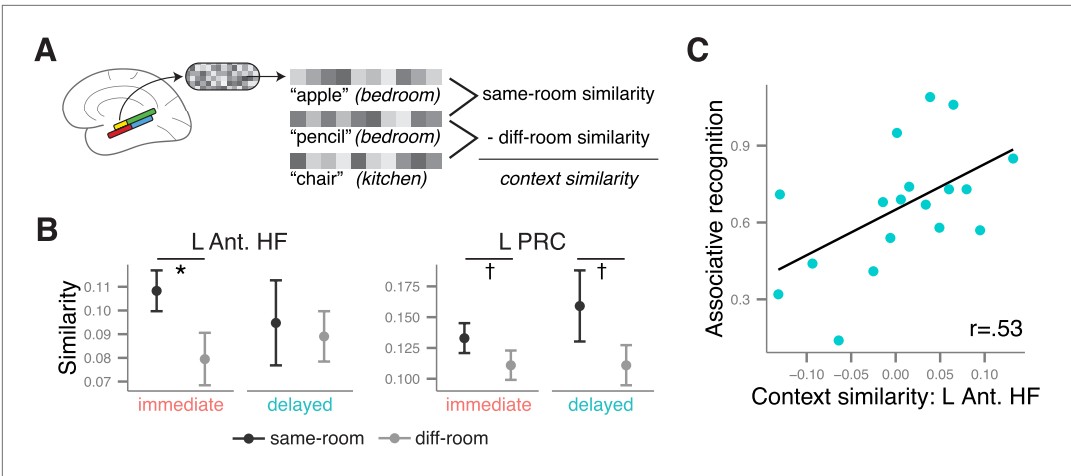

**Figure 4**. Context similarity in anterior MTL during recollection. (**A**) Schematic of the pattern similarity analysis procedure. (**B**) Estimates of pattern similarity (Pearson's r) for same-room and different-room pairs are plotted for the left anterior HF and left PRC (see *Figure 4—figure supplement 1* for other regions). Asterisk (*) denotes a significant effect of context similarity, that is, the difference in similarity for same- and different-room pairs, p < 0.05. Error bars denote the standard error of the mean. The cross (†) denotes a marginally significant effect, p < 0.08. Summary statistics for individual subjects are contained in *Figure 4—source data 1*. A non-parametric randomization test confirmed that same-room similarity was greater than what was likely to be observed by chance (*Figure 4—figure supplement 2*). Furthermore, these effects were observed only for recollection trials, not familiarity trials (*Figure 4—figure supplement 3*). (**C**) The relationship between associative recognition accuracy (d') and context similarity in the left anterior HF.

The following source data and figure supplements are available for figure 4:

**Source data 1**. Pattern similarity estimates.

**Figure supplement 1**. Context similarity effects in all MTL ROIs.

**Figure supplement 2**. Randomization test confirming context similarity effects in the anterior MTL.

**Figure supplement 3**. Context similarity in the anterior MTL during familiarity.

context similarity was defined as the difference in pattern similarity between same-room and different-room pairs, evaluated separately for each ROI. Note that because the locations were typical rooms in a house, this form of context retrieval may reflect a mixture of both spatial and semantic context information (i.e., remembering the general location or semantic features of the room associated with the item).

Because only a few studies (*Staresina et al., 2012*; *Hannula et al., 2013*; *Hsieh et al., 2014*) have shown that MTL multi-voxel patterns carry information about previously learned context information, even at immediate recall, our first analyses sought to establish the presence of pattern information related to context similarity during immediate recollection, when context memory was strongest. In the left anterior HF, pattern similarity was significantly greater among same-room pairs than different-room pairs, t(18) = 2.34, p = 0.015 (*Figure 4B*), and a similar trend was observed in the left PRC, t(18) = 1.73, p = 0.051. There were no significant effects for any of the other MTL regions, ts < 1, ps > 0.18 (*Figure 4—figure supplement 1*). Context similarity effects were also absent from cortical regions outside of the MTL, either immediately or after a delay, ts < 1.8, ps > 0.05, suggesting that these effects were selective to the left anterior MTL. The MTL findings were verified with a randomization test showing that pattern similarity for same-room pairs exceeded what would be expected by chance if context information were randomly assigned (*Figure 4—figure supplement 2*). Additionally, when the same analysis was run with familiarity trials instead of recollection trials, no differences were observed between same and different context trial pairs, ts < 1, ps > 0.2 (*Figure 4—figure supplement 3*). This distinction is consistent with the claim that pattern similarity differences were related

to the reactivation of shared context information, which should be more evident during memory recollection.

After determining that left anterior MTL regions showed evidence for context similarity at the immediate test, we next tested whether context similarity effects in these regions changed over time. The left PRC showed a significant main effect of context similarity across both delays, $F(1, 18) = 5.78$, $p = 0.027$, and context similarity did not interact with delay, $F(1, 18) = 0.54$, $p = 0.47$. In fact, the PRC showed a marginal context similarity effect for the delayed list as well, $t(18) = 1.54$, $p = 0.07$, suggesting that the sensitivity of this region to context information was stable over time. The anterior HF, on the other hand, showed neither a significant main effect, $F(1, 18) = 2.99$, $p = 0.10$, nor interaction, $F(1, 18) = 1.13$, $p = 0.30$. The absence of a clear main effect or interaction may be due, in part, to substantial variability in context similarity effects during delayed recollection. One possibility is that this variability is related to individual differences in retention of context information in memory. To test this hypothesis, we correlated individual subjects' context similarity estimates (i.e., the voxel pattern differences between same- and different-context trial pairs) with their performance on the subsequent associative recognition test, based on the idea that the associative recognition test relied on the same kind of room information measured by the pattern similarity analysis. Indeed, participants who performed best on the associative recognition test also showed the largest context similarity effect in the left anterior HF during delayed recollection, $r = 0.53$, $t(16) = 2.53$, $p = 0.022$ (*Figure 4C*). This relationship was significant even after controlling for individual differences in the number of recollection trials, $t(15) = 2.81$, $p = 0.013$. For the PRC, the correlation was in the same direction but not significant, $r = 0.25$, $t(16) = 1.02$, $p = 0.32$. Finally, there was no significant relationship between anterior HF context similarity and associative recognition for the immediate list, $r = -0.17$, $t(16) = 0.71$, $p = 0.49$; however, there was little variability in the similarity estimates for these trials. These findings provide strong evidence that, after a delay, the context similarity analysis was picking up on meaningful associative information during the item recollection period. Moreover, they are consistent with the memory transformation account, in that anterior HF involvement in context coding over time was contingent on the retention of context information in memory.

## Discussion

The current study provided novel evidence that HF and parahippocampal areas play different roles in supporting memory over time. During item recollection, the involvement of cortical MTL areas was stable over time. Within the HF, however, there were different effects for anterior and posterior regions. Whereas posterior HF showed reduced recollection-related activity after a 1-day delay, recollection-related activity in anterior HF remained relatively stable. Furthermore, anterior HF activity patterns carried information about contexts associated with the items at study, and this pattern information was maintained over time in participants who successfully retained context associations. Below, we relate these findings to extant neurobiological models of memory and discuss the specific contributions of individual MTL regions. We conclude that models of memory must account for variability among MTL subregions with respect to their involvement in memory retrieval over time.

### Differences in recollection-related activity along the longitudinal axis of the HF

Both the standard consolidation model and memory transformation accounts predict that HF activity during retrieval should decline with increasing retention intervals, although they assign different explanations to this decline. To our knowledge, neither account makes explicit predictions about variability along the longitudinal axis of the HF. In the present study, the anterior and posterior HF had dissociable responses during immediate and delayed retrieval. Whereas recollection-related activity in the posterior HF decreased over time, activity in the anterior HF was relatively stable across the 1-day delay. In previous fMRI studies that used paradigms comparable to the one used here, HF results have been somewhat mixed: some studies have shown that HF activity was greater for early than delayed retrieval (*Takashima et al., 2006*; *Sterpenich et al., 2009*; *Takashima et al., 2009*; *Yamashita et al., 2009*; *Milton et al., 2011*; *Watanabe et al., 2012*), but others have reported no change (*Stark and Squire, 2000*; *Janzen et al., 2008*; *Suchan et al., 2008*) or even the opposite effect (*Bosshardt et al., 2005b*; *Gais et al., 2007*). Some of these differences may be related to the sensitivity of the fMRI analysis to specific memory processes: some studies reporting no change used simple comparisons of targets and foils (*Stark and Squire, 2000*) or recognized and forgotten trials (*Janzen et al., 2008*),

whereas studies that isolated activity for high-confidence, recollection-based or associative hits have often reported delay-dependence (*Takashima et al., 2006*; *Sterpenich et al., 2009*; *Takashima et al., 2009*; *Yamashita et al., 2009*; *Milton et al., 2011*; *Watanabe et al., 2012*; but see *Suchan et al., 2008*). Although some delay effects have been localized to the anterior HF (*Takashima et al., 2006*; *Milton et al., 2011*), differences in the posterior HF have been commonly observed for studies using associative memory tasks (*Takashima et al., 2009*; *Yamashita et al., 2009*; *Watanabe et al., 2012*). By investigating delay effects within anatomically restricted regions, the current approach might have improved our ability to detect differences in localization that might not have been readily apparent in a group analysis applying voxel-wise thresholds. Thus, the finding that changes in recollection-related activity were circumscribed to the posterior HF is consistent with the available literature, but would not have been predicted by extant models.

Studies of autobiographical memory have examined differences between recent and remote memories across a more extended timescale (*Cabeza and St Jacques, 2007*), and some of these studies have compared the role of the anterior and posterior HF. For example, one study reported that the anterior HF showed a larger delay-dependent difference than the posterior HF during autobiographical memory retrieval (*Gilboa et al., 2004*). Another study used multi-voxel pattern analysis to decode information about autobiographical events. This study reported that patterns in the posterior HF could be used to decode remote but not recent autobiographical memories during repeated retrieval events, whereas patterns in the anterior HF could be used to decode both recent and remote memories (*Bonnici et al., 2012a*, *2013*). At face value, these findings might seem to contradict the present results, but there are numerous differences between autobiographical memory tasks and paradigms that focus on laboratory-controlled events. In particular, the use of repeated retrieval events and long retention delays could complicate the interpretation of fMRI studies of autobiographical memory retrieval. During remote autobiographical retrieval, hippocampal activity could be related to the construction of a new memory for an old event, or to the retrieval of the original memory or its more recent reconstructions. Another issue is that differences between recent and remote autobiographical memories can be confounded by differences in vividness and context specificity. Indeed, in the study by *Gilboa et al. (2004)*, there were no significant delay-dependent differences in the HF after controlling for vividness. Although laboratory-controlled studies are not immune to this possible confound, the issue was mitigated in the present study by limiting analyses to memory retrieval accompanied by recollection and by identifying neural patterns associated with studied context information.

An important question for future research is how changes in neural representation over a 1-day interval relate to the changes over longer intervals that are the focus of memory consolidation and transformation accounts. There is considerable evidence suggesting that early stages of systems consolidation can be initiated during the first night of sleep after learning (*Born and Wilhelm, 2012*), but little is known about how initial changes in hippocampal representation are related to longer-lasting changes. It is possible that the changes in activation magnitude and pattern similarity observed here reflect forgetting of certain aspects of context information, and that qualitatively different kinds of changes—particularly in cortical representation—might be apparent over a longer interval.

## Contributions of the anterior HF to context memory

The relative stability of recollection-related activity in the anterior HF may be related to its continued involvement in supporting context memory over time. Because recollection can be triggered by different kinds of associative details, including information about encoding context, we separately examined the sensitivity of the multi-voxel patterns to context information learned during encoding. Multi-voxel patterns in the left anterior HF were sensitive to context similarity during immediate recollection, and after a delay this effect was strongest for participants who could accurately retrieve context associations. According to one view of HF function, the anterior HF may be especially involved in representing episodic context at a global level (*Poppenk et al., 2013*). In support of this view, the anterior HF contains cells with larger place fields than the posterior HF (*Kjelstrup et al., 2008*), and it is involved in coding information about the coarse location of objects (*Evensmoen et al., 2014*), the global position of landmarks (*Ekstrom et al., 2011*; *Morgan et al., 2011*), and the gist of memories (*Poppenk et al., 2008*; *Gutchess and Schacter, 2012*). Taking this view into consideration, it may be that pattern similarity effects in the anterior HF reflect the activation of generalized context information (e.g., general features of the room associated with the item at study) that was shared across many encoding events. As discussed below, the posterior HF might support more specific representations

of context (e.g., remembering the exact location in which a specific item was imagined in the bedroom) in a more time-limited way. Regardless of the scale of anterior HF representations, it is noteworthy that recollection-related activity in the anterior HF was relatively stable over the delay, and that the anterior HF continued to show evidence for context coding in participants who retained context information in memory, suggesting that the type of information carried by the anterior HF can be long lasting. These findings are compatible with the memory transformation account, which models posit that the HF should be involved in retrieval so long as its preferred form of mnemonic information is maintained.

Irrespective of delay, an important finding of the present study was that multi-voxel patterns carried information about incidentally reactivated context associations during item recollection. A few previous studies have shown that neural patterns present during encoding are reactivated during item recognition (*Johnson et al., 2009*; *Ritchey et al., 2013*) and cued recall (*Staresina et al., 2012*; *Kuhl and Chun, 2014*), and that patterns within the MTL carry information related to study context (*Hannula et al., 2013*; *Hsieh et al., 2014*). For instance, during cued recall, patterns in the anterior HF, PRC, and PHC were shown to carry information about the study context (*Hannula et al., 2013*). However, there has been little evidence that item recollection involves the spontaneous reactivation of context-related patterns even without an overt source decision, despite the assumption that recollection typically involves contextual retrieval. In one study, *Johnson et al. (2009)* showed that a classifier trained to discriminate among study contexts was sensitive to context information that was incidentally reactivated during recognition, and that for the posterior cingulate, context reactivation was especially apparent for items accompanied by recollection. However, this study did not report context reactivation effects within the HF. The present work expands on these findings to demonstrate that, during recollection, left anterior HF patterns carried information about the context associated with an item during encoding, even when context was not explicitly cued or re-presented. The context similarity effect in the HF was relatively small, perhaps because HF patterns are more sensitive to similarities in object–context associations than to context alone (*Hsieh et al., 2014*; *Libby et al., 2014*). The finding that context-related pattern similarity was predictive of associative recognition performance, however, provides converging evidence that hippocampal voxel patterns carry behaviorally relevant context information. Nevertheless, future work will be necessary to determine the sensitivity of hippocampal voxel pattern information to other kinds of context manipulations.

## Similarities between the anterior HF and PRC

The PRC had a similar response profile to the anterior HF, in that it was stable in its recollection-related activity over time. It also tended to show context similarity effects that persisted across the delay. The anterior HF is strongly connected with the PRC, which is part of an anterior temporal system thought to be important for processing, remembering, and assigning value to items (*Ranganath and Ritchey, 2012*). These two regions may work together to support memory for item or item–context associations, and context similarity effects in the anterior HF may reflect the retrieval of contextual information in response to an item cue. The PRC, on the other hand, has been linked to item recognition (e.g., [*Davachi et al., 2003*; *Ranganath et al., 2003*]), recollection of item associations (*Diana et al., 2010*), and semantic processing (e.g., [*Wang et al., 2010*; *Clarke and Tyler, 2014*]). Thus, the prior literature is most consistent with a role for the PRC in item processing, and the finding of a context similarity effect in the PRC was unexpected. However, there is some evidence that the PRC may additionally carry some information about context, such as the locations of items in space (*Burke and Barnes, 2014*). PRC lesions have been shown to disrupt some forms of context memory, including object–context associations (*Norman and Eacott, 2005*), positional changes in object arrays (*Norman and Eacott, 2005*), and contextual fear (*Bucci et al., 2002*)—although these impairments have typically been more circumscribed than those observed following PHC damage (*Norman and Eacott, 2005*). Based on this literature and the present data, we cannot rule out the possibility that like the anterior HF, the PRC is involved in the long-term storage of the association between an object and the general context in which it was encountered. Alternatively, it could be that the anterior HF and PRC are both sensitive to shared context information, but for different reasons: whereas the anterior HF might carry general representations of context, pattern similarity effects in the PRC might reflect the recollection of items associated with each room, through episodic associations learned during encoding or through existing semantic associations.

## The time-limited role of posterior HF in recollection

In contrast to the anterior HF and PRC, the posterior HF was neither stable in its contributions to recollection over time nor did it show enhanced pattern similarity during retrieval of objects that shared the same contextual associations. As noted above, it is possible that the posterior HF encodes highly specific contextual details (*Poppenk et al., 2013*), such as precise locations within a spatial context (*Kjelstrup et al., 2008*; *Evensmoen et al., 2014*) or positions within a sequence (*Hsieh et al., 2014*), which might be useful for disambiguating among related contexts. For instance, in one study, hemodynamic responses in the posterior HF were greater during the retrieval of precise rather than coarse location information associated with an object (*Evensmoen et al., 2014*). It is possible that, here, the reduction in posterior HF activity across the delay reflects the forgetting of trial-specific context information. Prior work in rodents has shown that memories may lose their contextual specificity over time (*Wiltgen and Silva, 2007*; *Winocur et al., 2007*), and that the loss of specificity is associated with diminished dependence on the dorsal HF (which may be homologous to the posterior HF in humans) during retrieval (*Wiltgen et al., 2010*). In addition, if the posterior HF codes only specific context information, this could explain the absence of a context similarity effect in this region. By definition, the multi-voxel pattern similarity analysis used here depended on similarities among trials that had been associated with the same 'room', and was therefore insensitive to trial-specific information. Some previous studies have used approaches that have enabled them to identify event- or scene-specific patterns, and these studies have shown that this kind of specific information can be decoded from multi-voxel patterns in the HF (*Chadwick et al., 2011*; *Bonnici et al., 2012b*; *Hsieh et al., 2014*), and that the specificity of HF patterns is positively related to memory performance (*LaRocque et al., 2013*). In this study, we were unable to directly measure trial-specific information in the posterior HF, but future studies could incorporate graded levels of context specificity in order to test its relation to response changes over time.

## Neocortical contributions to recollection

The role of the MTL in memory has often been contrasted against the role of neocortical areas in memory, which are thought to increase or remain stable in their support of memory over time (*Squire, 1992*; *Nadel and Moscovitch, 1997*; *Winocur et al., 2010*). However, most models have excluded the PRC and PHC from their discussion of cortical function ([*Squire, 1992*; *Nadel and Moscovitch, 1997*; *Winocur et al., 2010*; but see *Norman and O'Reilly, 2003*) and previous studies have not typically included these regions as ROIs (but see *Stark and Squire, 2000*). Here, we provide novel evidence that the roles of the HF and parahippocampal areas are dissociable in terms of their involvement in recognition memory over time. Whereas recollection-related activity in the posterior HF declined from immediate to delayed retrieval, activity in the PRC and PHC remained stable. One interesting point of divergence is between the posterior HF and PHC, which are strongly interconnected (*Ranganath and Ritchey, 2012*). Like the posterior HF, the PHC has been implicated in memory for specific context information (*Davachi, 2006*; *Eichenbaum et al., 2007*; *Ranganath, 2010*), and the PHC has also been shown to carry information about scene-specific associations (e.g., *Staresina et al., 2012*). Here, patterns in both regions were insensitive to shared context information, yet the PHC was involved in memory recollection after a 1-day delay, whereas the posterior HF was not. Future work should address whether PHC representations simply remain more stable over time, as compared with those in the posterior HF, or whether there are other differences that might explain their different activation profiles.

Beyond the MTL, across both delays, recollection-related activity was also observed in a network of regions including the retrosplenial cortex, posterior cingulate, precuneus, angular gyrus, and medial prefrontal cortex. This network is thought to be important for context memory and recollection (*Bar, 2004*; *Ranganath and Ritchey, 2012*; *Rugg and Vilberg, 2013*). Importantly, these findings suggest that most areas involved in recollection, whether they are within the MTL or not, maintain their involvement across a 1-day delay. Altogether, these findings clearly demonstrate that neurobiological models of memory must go beyond simple dichotomies between the MTL and neocortex to address the role of specific HF and parahippocampal areas.

## Future directions and conclusions

The present results raise several questions to be addressed in future studies. One question involves the role of the PRC and PHC in immediate and delayed recollection. Here, we found that both regions

supported recollection immediately and after a 1-day delay, but that patterns in the PRC but not the PHC were sensitive to shared context information. Future work should address whether the PRC and PHC are similarly involved in memory tested after longer intervals and with measures that are more sensitive to the specificity of information retained in memory. Additionally, because memory was tested after a 1-day delay, we cannot disentangle changes attributable to active sleep-dependent processes (*Ellenbogen et al., 2006*) from more passive time-dependent changes. Finally, an important next step would be to compare the item–location associations used here with associations that may be less dependent on HF function, such as unitized associations (*Giovanello et al., 2006*; *Quamme et al., 2007*) or emotional associations. In particular, emotional memories are forgotten more slowly than neutral memories (e.g., *Sharot and Yonelinas, 2008*), and some evidence suggests that the persistence of emotional recollection is related to the function of anterior MTL structures (*Ritchey et al., 2008*).

In summary, this study provided novel evidence that regions within the MTL play different roles in supporting item recollection over time. The results highlight the need to revise existing models to incorporate differences between MTL areas. In particular, it will be important for models to distinguish between the anterior and posterior HF, which may show different changes in their contributions to memory over time.

## Materials and methods

### Participants

Data were acquired from 30 young adults (15 female; ages 18–31 years). Data from one participant was excluded due to button box issues during the scan, and data from another participant was excluded due to head motion and poor performance. Of the remaining 28 participants, 9 were excluded from fMRI data analyses due to insufficient variability in memory performance (i.e., fewer than 9 recollection or familiarity trials). Thus, the fMRI analyses included 19 participants (9 female). Due to technical problems, one of these participants completed the item recognition but not associative recognition task. Participants reported that they were native English speakers, free of neurological and psychiatric disorders, and eligible for MRI. Participants reported sleeping, on average, 7.42 hr (range: 5–12 hr) between the first and second session.

### Stimuli

Stimuli consisted of 252 nouns that referred to objects. For each participant, these words were randomly assigned to one of three lists (N = 84 each): the Day 1 encoding list (delayed list), the Day 2 encoding list (immediate list), or the lure list for item recognition. During encoding, items were placed into sentences describing the location of the object, which could be in one of eight rooms in a house: bathroom, bedroom, den, dining room, kitchen, living room, office, and patio area. For example, on one trial, a possible sentence might read, 'The apple is in the bedroom'. Thus, the room associated with each item constituted its encoding context, which might include a mixture of spatial and semantic information about the room. The eight contexts were randomly assigned to either the immediate or delayed encoding list, such that only four contexts were presented on either day.

### Experimental Design

There were two experimental sessions that occurred on consecutive days (*Figure 1A*). On Day 1, participants completed an encoding task. On Day 2, participants completed another encoding task, an item recognition task, and an associative recognition task. Both encoding tasks took place in the same laboratory testing rooms. The recognition tasks took place in the scanner, with the item recognition task beginning as soon as the participant was positioned within the scanner, approximately 20 min after the end of the Day 2 encoding task. All fMRI analyses focus on the item recognition task.

During the two encoding tasks, participants studied sentences in which trial-unique object nouns were paired with one of eight contexts (*Figure 1A*). The set of items and contexts was different on each day. Different contexts were assigned to each day in order to avoid confounds related to contextual interference across days. During each encoding task, 84 sentences appeared on-screen for 5 s each, separated by jittered fixation intervals (mean = 4 s, range = 2–10 s). Participants were instructed to rate on a continuum how well they were able to imagine the pairing on a 6-point scale, with 1 = *not well* and 6 = *very well*. Trial order was randomly determined for each participant. Immediately after each encoding task, participants were cued to group the four previously-studied rooms into two

houses, based on random assignment. This grouping manipulation did not alter memory performance and will not be considered further.

The item recognition task was designed to assess memory for the items studied during encoding. During this task, participants were presented with words from both encoding lists and the lure list (*Figure 1A*). Words were presented for 2 s each, separated by jittered fixation intervals (mean = 4 s, range = 2–10 s). Participants were asked to determine whether the word was old (studied either day) or new (unstudied) using a modified 6-point remember-know scale, including responses for *definitely new*, *probably new*, *not sure*, *probably old*, *definitely old*, and *remember*. For half of the participants, the scale was presented in reverse order. Participants were instructed that they should use the 'remember' response any time they could recall any kind of specific detail from when they initially studied that item, whereas the other memory responses reflected graded levels of memory confidence in the absence of a specific detail. We did not explicitly instruct the participants to remember the associated room during the item recognition phase; rather we emphasized that any type of detail would qualify for a 'remember' response. This was because we did not want participants to engage in a strategy in which they called to mind the rooms on every trial, which would have interfered with our ability to detect room information arising from memory. Trials from each list were evenly divided across three functional imaging runs, such that each run contained the same number of trials associated with each studied location. A unique sequence of trials and jittered fixation intervals was randomly determined for each participant.

The associative recognition task was designed to assess memory for the item–context associations made during encoding. During this task, participants were presented with sentences that were either identical to sentences that they had studied during either encoding session ('intact') or sentences that were recombinations of items and contexts that were both previously studied but not as part of the same sentence ('recombined'). Of the 84 sentences studied on each day, 28 were presented as intact and 56 were presented as recombined. The items and contexts in the recombined sentences were always drawn from the same day list. Sentences were presented for 3 s each, separated by jittered fixation intervals (mean = 4 s, range = 2–10 s). Participants were asked to rate whether the sentence was intact or recombined on a 6-point scale, including response for *definitely recombined*, *probably recombined*, *guess recombined*, *guess intact*, *probably intact*, and *definitely intact*. For half of the participants, the scale was presented in reverse order. Trials from each list were evenly divided across three functional imaging runs, and trial order was randomly determined for each participant.

## Behavioral analysis

Behavioral analyses were based on the full available sample for each task. Item recognition performance was measured as the discriminability (d') between old items (from the immediate or delayed list) and new items. Item recognition was further broken down into estimates of recollection and familiarity according to the dual-process model of recognition memory. Recollection was defined as $(R_{old} - R_{new})/(1 - R_{new})$, where $R_{old}$ is the rate of 'R' responses to old items, and $R_{new}$ is the rate of 'R' responses to new items. Familiarity was defined as $(F_{old}/(1 - R_{old})) - (F_{new}/(1 - F_{new}))$, where $F_{old}$ is the rate of 'definitely old' and 'probably old' responses to old items, and $F_{new}$ is the rate of 'definitely old' and 'probably old' responses to new items. Note that these process estimates were computed to allow comparison to previous studies. The primary findings of the study, however, do not depend on assumptions specific to the dual process model. Associative recognition performance was measured as the discriminability (d') between intact and recombined sentences, separately for the immediate or delayed encoding list. To determine the relation between item recognition and associative recognition, the proportion of correct associative recognition responses was calculated for items previously marked as recollected, familiar, or forgotten. All statistical comparisons on the behavioral data were conducted in R version 3.1.1 (http://www.R-project.org).

## Image acquisition and pre-processing

Scanning was performed on a Siemens Skyra 3T scanner system with a 32-channel head coil. High-resolution T1-weighted structural images were acquired using a magnetization prepared rapid acquisition gradient echo (MPRAGE) pulse sequence (field of view = 25.6 cm, image matrix = 256 × 256, 208 axial slices with 1.0 mm thickness). Functional images were acquired using a multi-band gradient echo planar imaging (EPI) sequence (TR = 1220 ms; TE = 24 ms; FOV = 19.2 cm; image matrix = 64 × 64; flip angle = 67; multi-band factor = 2; 38 axial slices; voxel size = 3.0 × 3.0 × 3.0 mm).

SPM8 (http://www.fil.ion.ucl.ac.uk/spm/software/spm8/) was used to pre-process the images, including realignment, normalization, and smoothing. The high-resolution T1 image was skull-stripped via segmentation. Functional images were realigned, correcting for motion, and resliced. Resliced, native-space images served as the basis for the anatomical ROI analyses, in which manually segmented ROIs (see *ROI Definition*) were co-registered to the mean functional. For group voxel-wise analyses, the mean functional was co-registered to the skull-stripped anatomical image, moving all of the functional images in register with the anatomical image. At this point, the anatomical and functional images were warped to a group-derived template generated using diffeomorphic registration (DARTEL) and normalized to MNI space. Functional images were smoothed with a 6-mm Gaussian kernel. Skull-stripped anatomical images were also warped and smoothed for use as an explicit mask for subsequent functional analyses. Quality assurance included the identification of 'suspect' time-points via the Artifact Detection Tools (ART; http://www.nitrc.org/projects/artifact_detect), defined as time-points marked by greater than 0.3 mm in movement or 1.3% global mean signal change. One participant was excluded from analysis due to excess motion (>3 mm) within the functional runs.

## ROI definition

The anterior HF (HF head), posterior HF (HF body and tail), PRC, and PHC were manually segmented on the MPRAGE coronal plane according to previously published guidelines (*Insausti et al., 1998*; *Franko et al., 2014*). In brief, the most posterior slice of the anterior HF was defined as the last slice containing the gyrus intralimbicus; the posterior HF immediately followed. The anterior extent of the PRC was defined as 2 mm anterior to the limen insula or the most anterior slice in which the collateral sulcus was visible, whichever was more anterior. The most posterior slice of the PRC was defined as 4 mm posterior to the anterior/posterior HF transition. The PHC immediately followed the PRC, and the posterior extent was defined as 2 mm posterior to the appearance of the posterior crus of the fornix. The PRC segmentation included the entire lateral bank and dorsal half of the medial bank of the collateral sulcus. The PHC segmentation included the medial bank of the collateral sulcus, extending to the most medial aspect of the parahippocampal gyrus. Some analyses also included a set of regions outside of the medial temporal lobes, including the retrosplenial cortex, posterior cingulate, precuneus, angular gyrus, and medial prefrontal cortex. These ROIs were labeled with FreeSurfer cortical parcellation tools (http://surfer.nmr.mgh.harvard.edu/) using the Destrieux atlas (*Destrieux et al., 2010*) for the following labels: G_cingul-Post-ventral, G_cingul-Post-dorsal, G_precuneus, G_pariet_inf-Angular, and S_suborbital. Segmented brains were co-registered to the mean functional image and split into masks for each ROI (see *Figure 1B* for an example set of MTL ROIs). Masks were filtered to exclude voxels with low signal, defined as having mean temporal SNR (calculated across all functional runs) more than 1 standard deviation below the ROI mean. For visualization in standard space, ROI masks were warped to MNI space and combined across subjects into probabilistic maps.

## Data analysis

For ROI analyses, models were run on unsmoothed functional images in native space. ROI summary statistics, including pattern similarity estimates, were extracted with in-house scripts (*Source code 1*) in MATLAB 2009b (The MathWorks, Inc., Natick, MA), and statistical comparisons were conducted in R version 3.1.1 (http://www.R-project.org). For voxel-wise analyses, models were run on smoothed functional images in standard MNI space, and statistical comparisons were conducted in SPM8.

### Univariate activation analyses

Event-related stick-function regressors were used to model trials corresponding to one of nine conditions: immediate recollection, immediate familiarity, immediate forgotten, delayed recollection, delayed familiarity, delayed forgotten, correct rejections, false alarms, and no-response trials. 'Recollection' trials were defined as old items that were correctly recognized and endorsed with recollection (i.e., *R* response). 'Familiarity' trials were defined as old items that were correctly recognized but not endorsed with recollection (i.e., *probably old* or *definitely old* responses). Six motion parameter regressors were included in the model. Spike regressors were also included to model time-points identified as ART suspects. Whole-brain fixed-effects contrasts were evaluated to obtain estimates of activity in response to each trial type relative to implicit baseline. Contrast maps for 'recollection-related activity' were created by computing the activation difference between recollection and familiarity trials, separately for each delay. The difference in recollection-related activity contrast images

between immediate and delayed trials was then used to estimate delay-dependent changes in recollection-related activity.

For ROI random-effects analyses, contrast estimates were averaged within ROI masks of the left and right anterior HF, posterior HF, PRC, and PHC. Contrast means were compared with repeated-measures ANOVAs with factors for memory status (recollection, familiarity) and delay (immediate, delayed). For comparison of effects across ROIs, estimates of recollection-related activity were compared with repeated-measures ANOVAs with factors for ROI, hemisphere, and delay. For completeness, activity estimates for familiarity and miss trials were also compared, but recollection-related activity was the a priori focus of the experiment. For voxel-wise random-effects analyses, contrast maps were evaluated with one-sample t-tests. Clusters were considered significant (cluster-corrected $p < 0.05$) if they contained at least 36 voxels within a mask of the entire brain, based on simulations with the 3dClustSim tool in AFNI (http://afni.nimh.nih.gov). Delay-insensitive effects were identified as clusters that showed recollection-related activity for both immediate and delayed lists, each thresholded at $p < 0.032$ for a joint voxel-wise threshold of $p < 0.001$, exclusively masking for any significant effect of delay (liberally defined at $p < 0.05$).

## Pattern similarity analysis

Pattern similarity analyses (*Kriegeskorte, 2008*) were conducted on unsmoothed functional images in native space. Single trial models were generated to estimate the response to each individual trial (N = 252 per participant), resulting in a beta image for every trial. Similar to the procedure described by *Mumford et al. (2012)*, a separate general linear model was run for each individual trial, with the first regressor containing a stick function mapped to the onset of the individual trial and the second regressor containing stick functions modeling all of the other trials, with additional motion and nuisance regressors as described above. For each participant, the voxel-wise pattern of hemodynamic activity within each ROI was extracted from each of the 252 single-trial beta images.

Separately for each ROI, trial patterns were correlated with each other using Pearson's r. Correlations were limited to pairs of trials from the same encoding list and memory status: that is, immediate recollection trials were correlated only with other immediate recollection trials (and likewise for delayed trials). Pattern similarity values were aggregated according to whether or not the items had shared context information during encoding, that is, same-room vs different-room similarity. For instance, as depicted in *Figure 4A*, the item pair 'apple' and 'pencil' share information because they were encoded in sentences pairing them with the same room ('bedroom'), whereas the item pair 'apple' and 'chair' do not share information because they had been paired with different rooms. Different-room pairs excluded pairs of rooms that had been experimentally grouped after encoding (see 'Experimental design'). Thus, for any given item, its same-room pairs included all other items encoded with the same room, and its different-room pairs included items encoded with one of two other rooms on the same day. To control for differences in pattern similarity between runs, same-room and different-room similarity were first averaged within each run, and then similarity values from the three runs were averaged together. To control for similarities among adjacent trials, only pairs at least two trials apart were included in the analysis; however, because trial sequences were randomly determined for each participant, temporal autocorrelation was unlikely to produce spurious pattern similarity effects at the group level (*Mumford et al., 2014*). Pairs containing trials with outlying global signal values were also excluded, based on the global average of absolute standardized values calculated for each within-brain voxel. Pattern similarity values were Fisher-transformed for statistical comparison.

Because room information was not present during the item recognition phase and must be attributed to memory processes, the difference between same-room and different-room similarity was taken as evidence for context similarity. This difference was tested with one-sample *t*-tests based on our directional hypothesis that same-room similarity should be greater than different-room similarity. For regions showing evidence for context similarity in the immediate condition, when context memory was strongest, we additionally tested the influence of delay on context similarity was tested with repeated-measures ANOVAs with factors for pair type (same-room, different-room) and delay (immediate, delayed). Finally, Pearson's correlation was used to test the relationship between context similarity and subsequent associative recognition performance for the delayed list, and multiple linear regression was used to verify that the observed relationship was not explained by differences in the number of item recollection trials. One participant was excluded from the regression analyses due to below-chance associative recognition performance.

## Acknowledgements

This research was supported by NIMH R01 MH083734 (CR and APY), a Guggenheim Fellowship (CR), and a Leverhulme Trust Visiting Professorship (CR). We would like to thank Laura A Libby, Priyanka Patel, and Deepika Dokuru for advice and assistance related to ROI segmentation, Aneil Dhillon for assistance with data collection, Shao-Fang Wang for assistance with figure preparation, and members of the UC Davis Memory Group for general advice and comments.

## Additional information

### Funding

| Funder | Grant reference number | Author |
| --- | --- | --- |
| National Institute of Mental Health | R01MH083734 | Andrew P Yonelinas, Charan Ranganath |
| John Simon Guggenheim Memorial Foundation | | Charan Ranganath |
| Leverhulme Trust | | Charan Ranganath |

The funders had no role in study design, data collection and interpretation, or the decision to submit the work for publication.

### Author contributions

MR, Contributed new analytic tools, Conception and design, Acquisition of data, Analysis and interpretation of data, Drafting or revising the article; MEM, Contributed new analytic tools, Acquisition of data; APY, CR, Conception and design, Drafting or revising the article

### Author ORCIDs

Maureen Ritchey, http://orcid.org/0000-0002-5957-3642
Charan Ranganath, http://orcid.org/0000-0001-5835-6091

### Ethics

Human subjects: The study was approved by the Institutional Review Board at the University of California, Davis (protocol #238604). Written informed consent was obtained from each participant before the experiment, including consent to publish anonymized results.

## Additional files

### Supplementary files

• Supplementary file 1. MTL activity estimates for all conditions (related to *Figure 2*).

• Supplementary file 2. Table of peak coordinates from the voxel-wise comparison of recollection and familiarity trial activity (related to *Figure 3*).

• Source code 1. Custom Matlab code for the pattern similarity analysis.

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
