## [Decision Letter]

Thank you for sending your work entitled “Delay-dependent contributions of medial temporal lobe regions to episodic memory retrieval” for consideration at *eLife*. Your article has been favorably evaluated by K VijayRaghavan (Senior editor), a Reviewing editor, and 3 reviewers.

The following individuals responsible for the peer review of your submission have agreed to reveal their identity: Howard Eichenbaum (Reviewing editor) and Robert Komorowski (peer reviewer). Two other reviewers remain anonymous.

The Reviewing editor and the reviewers discussed their comments before we reached this decision, and the Reviewing editor has assembled the following comments to help you prepare a revised submission.

In general there was consensual enthusiasm for the findings, especially the observed distinction between anterior and posterior hippocampus and the implications for theories of consolidation. However, at the same time there were several concerns about the some of the comparisons, about the strength of some of the results, and about interpretation voiced by individual reviewers, whose comments are listed below.

Reviewer 1:

1) It has been shown that the reaction time of respondents differs between strong and weak memories with weaker memories having longer reaction times (i.e. Stark and Squire, 2000). The authors should provide and statistically compare reaction times in the immediate vs. delayed conditions across the various confidence levels and the same for the test of associative recognition. If there are differences, particularly between the immediate and delayed conditions for the key “remember” condition, it could be argued that the regional activation differences they see are due in part to differences in reaction time and not the memory delay. To address this possible issue the authors should select trials where reaction time overlaps between the immediate and delayed conditions and see if their main results persist.

2) Their analysis focuses on recollective-based recognition but largely ignores trials where recognition was based upon familiarity. Previous work has suggested that familiarity-based recognition is supported preferentially by MTL cortices. It would be of interest to ascertain whether there are delay-dependent differences in familiarity-based activity to determine whether consolidation processes in the MTL are limited to recollective recognition or occur for all new memories.

3) The pattern analysis of contextually related activity in the anterior hippocampus provides strong evidence for the specific contribution and stability of this region to recall of object/context associations. One wonders whether a similar analysis could be performed for object-related activity to determine which brain regions (possibly posterior hippocampus?) are involved in object processing and whether those regions change over time. Unfortunately, the analysis wouldn't be as elegant as the analysis for context which looks at context activation which only the word present, but assuming there are multiple recognition trials with same object/context pair, the authors could compare the average pattern similarity between trials with the identical object/context pairs versus pairs with two different objects in the same room. This would isolate coding reflective of the item only and reveal what brain regions correlate more for same object/context pairs and then see if their involvement of these changes over time.

4) Figure 4 plots the relationship between associative recognition and the context similarity observed in the anterior hippocampus. However, the authors also observe significant coding for context in the perirhinal cortex. The authors should also examine whether a similar relationship between context similarity and associative recognition exist in this region.

Reviewer 2:

1) Delay-invariance in the anterior hippocampus and neocortical areas: this seems to be an overly strong description of the actual pattern found here. There was a numerical decrease in recollection-related activity in Ant HF and most neocortical regions. Decrease in Left precuneus (Figure 3) is said not to reach significance, but the pattern is unlikely significantly different than what observed in posterior hippocampus. Right PCC and precuneus (Figure 3—figure supplement 1) actually decreased significantly. Visually, the effects for delayed recollection are overall just weaker than for immediate recollection (Figure 3 plus Figure 3–figure supplement 1–3). Two analyses are listed to support the “invariant” claim: The conjunction analysis (Figure 3–figure supplement 3) looked for voxels that are p<.032 for both delay and immediate condition (called joint p<.001), but that led to a larger “conjunctive map” than the maps related to either condition alone (compare Figure 3 with Figure 3–figure supplement 3). Therefore, it should be made clear this is not a map of clusters that were significantly recollection-related in immediate condition and also significantly recollection-related in delayed condition (as it may sound on the first read). Rather, these are voxels somewhat involved in both conditions, jointly reaching significance. If I read correctly, the authors also did direct contrast of recollection effects for immediate minus delay (“while there were no clusters showing significant delay differences (corrected p<.05)”, p. 8). This analysis/result seems a more straightforward way to support the invariance claim and should be stated more clearly. One caveat is that it also fails to find any clusters in posterior hippocampus, whose involvement is shown to decrease using the ROI approach.

2) Context similarity in anterior MTL (especially Ant HF) is a nice finding that carries a lot of weight. Ant HF represented context (associated room) for the immediate condition, and while it did not represent context for the delayed condition across the group, it appears to do so in subjects who did remember the associated room in the associative text. My main concern is that the finding is somewhat weak (authors used p < 0.05, one-tailed test, uncorrected, while making at least 8 tests (8 ROIs across left and right MTL)).

3) The authors also argue that their results pose some challenge to both standard consolidation model and memory transformation model because they would not predict the differences between anterior and posterior HF. In my view, the lack of subregional specificity in those models does not necessarily mean a challenge to them. Furthermore, most consolidation literature concerns longer delays than just one day (sometimes years). Therefore, the current experiment may be tapping into delay-related changes in recollection but it is not clear if they are related to consolidation. For instance, the authors note that Post HF may code for item-specific context (not measurable here). Post HF may no longer show recollection effects after a delay because this specific context was forgotten over the delay. All what is left may be more general context (such as the associated room) represented in Ant HIP, observable at least in the subjects who remembered the associated room.

Reviewer 3:

It would be helpful to have figures depicting the similarity results at immediate and delay timepoints for the PHC and post HF (which are reported to not show context similarity effects and/or effects that are sustained over time) alongside those presented for ant HF and L PRC as in Figure 4. It is one thing to note that the similarity effects are not present or stable, but the baseline level of similarity in PHC and post HF may be higher than in other regions, which may affect the conclusions regarding function. The conclusion that similarity in Ant HF is maintained over a delay does not receive strong support in the data considering that there is no difference between same, and different, room similarity in delayed recognition. This would then have consequences for the interpretation. The basis for this conclusion of stability rests on the correlation between similarity and recognition performance, but while this addresses the contributors to performance, it does not address stability per se. It would also be helpful to see whether the correlation between immediate similarity and recognition is also significant.

---

## [Author Response]

Reviewer 1:

*1) It has been shown that the reaction time of respondents differs between strong and weak memories with weaker memories having longer reaction times (i.e. Stark and Squire, 2000). The authors should provide and statistically compare reaction times in the immediate vs. delayed conditions across the various confidence levels and the same for the test of associative recognition. If there are differences, particularly between the immediate and delayed conditions for the key “remember” condition, it could be argued that the regional activation differences they see are due in part to differences in reaction time and not the memory delay. To address this possible issue the authors should select trials where reaction time overlaps between the immediate and delayed conditions and see if their main results persist*.

The reviewer expressed a concern that the difference in posterior hippocampal activation between the immediate and delayed recollection contrasts could be driven by differences in response times (RT). To be clear, our primary activation analysis did not compare immediate and delayed “Remember” trials. Instead, we tested for a delay-dependent difference between “Remember” and “Familiar” trials. Specifically, in the posterior hippocampus, there was an interaction between delay and response type, such that activity was greater for Remember than for Familiar trials for the immediate but not the delayed list. If RT confounds were responsible for the delay-dependent activation difference, then we would expect that RTs would also show an interaction between delay and memory response. We did not observe this interaction, F(1,26)=.06, p=.80. There was a main effect of delay, such that RTs were faster for all immediate items than for all delayed items, F(1,26)=13.87, p=.001, but this held true regardless of whether the item was recognized on the basis of recollection or familiarity. Note that this pattern also holds when considering only the participants included in the fMRI analyses (interaction: F(1,18)=.05, p=.83; main effect: F(1,18)=5.87, p=.026). Thus, response times are unlikely to account for why posterior hippocampal activation differences between Remember and Familiar trials were larger for immediate than delayed trials. We now include this RT analysis in the Results section of the revised manuscript.

Note that RT differences between immediate and delayed trials could not have affected the pattern similarity analyses, because for these analyses, comparisons were made among recollection trials pulled from the same day list (i.e., immediate recollection trials were paired with other immediate recollection trials, for both the same-room and different-room comparisons).

*2) Their analysis focuses on recollective-based recognition but largely ignores trials where recognition was based upon familiarity. Previous work has suggested that familiarity-based recognition is supported preferentially by MTL cortices. It would be of interest to ascertain whether there are delay-dependent differences in familiarity-based activity to determine whether consolidation processes in the MTL are limited to recollective recognition or occur for all new memories*.

The reviewer makes an excellent point that it is important to know whether there are parallel delay-dependent changes in activity related to familiarity and recollection-based recognition. Our a priori analyses focused on recollection-based recognition because we were interested in whether time-dependent changes in contextual retrieval could explain differences in retrieval-related neural activity over time. Many studies have shown that more contextual information is retrieved during “Remember” trials than during “Familiar” trials (e.g., Perfect et al., 1996, QJEP). Thus, we predicted that the comparison between recollection and familiarity trials would maximize sensitivity to detecting brain activity related to contextual retrieval.

Brain activity related to familiarity-based recognition is often assessed by contrasting activity between “Familiar” and “Miss” trials, but for the reasons described above, the experiment was designed to yield high levels of recollection. This focus on recollection may have reduced our sensitivity to detecting correlates of familiarity-based recognition due to a relatively small number of “miss” trials in the included sample (mean=12, minimum=4 for the immediate list). Perhaps as a result, only one of our MTL ROIs (the right posterior HF) showed a significant main effect of response type (familiarity > miss), F(1,18)=4.42, p=.049. None of the ROIs showed any sign of delay modulation of familiarity-based responses. This detail is now included in the Methods and Results sections. We have also added a table of mean activity estimates for all conditions in [Supplementary-material SD5-data].

For the pattern similarity results, no region showed an effect of context similarity when familiarity trials were considered, as reported in the Discussion section.

*3) The pattern analysis of contextually related activity in the anterior hippocampus provides strong evidence for the specific contribution and stability of this region to recall of object/context associations. One wonders whether a similar analysis could be performed for object-related activity to determine which brain regions (possibly posterior hippocampus?) are involved in object processing and whether those regions change over time. Unfortunately, the analysis wouldn't be as elegant as the analysis for context which looks at context activation which only the word present, but assuming there are multiple recognition trials with same object/context pair, the authors could compare the average pattern similarity between trials with the identical object/context pairs versus pairs with two different objects in the same room. This would isolate coding reflective of the item only and reveal what brain regions correlate more for same object/context pairs and then see if their involvement of these changes over time*.

We thank the reviewer for the suggestion and agree that it would be a nice way to investigate the neural patterns associated with item-specific information. However, it is not possible to run the suggested analysis in the current experiment, because items were trial-unique, i.e., each participant made only a single recognition decision per item.

*4)*
Figure 4
*plots the relationship between associative recognition and the context similarity observed in the anterior hippocampus. However, the authors also observe significant coding for context in the perirhinal cortex. The authors should also examine whether a similar relationship between context similarity and associative recognition exist in this region*.

We have now analyzed the relationship between context similarity in the left perirhinal cortex and associative recognition. The relationship was in the same direction, r=.25, as what we observed for the anterior hippocampus, but it did not reach significance, t(16)=1.02, p=.32. We now report this result in the Discussion section of the revised manuscript.

Reviewer 2:

*1) Delay-invariance in the anterior hippocampus and neocortical areas: this seems to be an overly strong description of the actual pattern found here. There was a numerical decrease in recollection-related activity in Ant HF and most neocortical regions. Decrease in Left precuneus (*Figure 3*) is said not to reach significance, but the pattern is unlikely significantly different than what observed in posterior hippocampus. Right PCC and precuneus (*Figure 3—figure supplement 1*) actually decreased significantly. Visually, the effects for delayed recollection are overall just weaker than for immediate recollection (*Figure 3
*plus Figure 3–figure supplement 1–3). Two analyses are listed to support the “invariant” claim: The conjunction analysis (Figure 3–figure supplement 3) looked for voxels that are p<.032 for both delay and immediate condition (called joint p<.001), but that led to a larger “conjunctive map” than the maps related to either condition alone (compare*
Figure 3
*with Figure 3–figure supplement 3). Therefore, it should be made clear this is not a map of clusters that were significantly recollection-related in immediate condition and also significantly recollection-related in delayed condition (as it may sound on the first read). Rather, these are voxels somewhat involved in both conditions, jointly reaching significance. If I read correctly, the authors also did direct contrast of recollection effects for immediate minus delay (“while there were no clusters showing significant delay differences (corrected p<.05)”, p. 8). This analysis/result seems a more straightforward way to support the invariance claim and should be stated more clearly. One caveat is that it also fails to find any clusters in posterior hippocampus, whose involvement is shown to decrease using the ROI approach*.

We thank the reviewer for the detailed and thoughtful comments. In our original submission, we used the term “delay-invariance” to refer to regions that did not significantly differ in their contributions to immediate and delayed recollection. However, we agree with the reviewer that some of the regions that we have referred to as “invariant” might show numerical differences that do not reach statistical significance, and thus “invariant” may be too strong of a term. We have revised the manuscript throughout to refer instead to relative “insensitivity” to the delay manipulation. We have also softened our claims about delay invariance in cortical regions outside of the MTL. In doing so, we point out the delay effects observed in the right precuneus and PCC (please see the Results section), although it is worth noting that these two regions did not strongly contribute to recollection in general.

We have also revised the manuscript to clarify that the conjunction shown in Figure 3—figure supplement 2 reflects a joint probability rather than each individual map reaching independent significance. It is a known problem that the application of statistical thresholds can set up situations where one contrast produces a “significant” effect where another does not, despite there being no true difference between the two contrasts; i.e., one contrast may be just above threshold and another may be just below threshold. Thus we feel that the most fair and complete way to present these data are to show the independent maps alone (as we do in Figure 3), as well as a conjunction that relaxes the individual thresholds while masking out any differences between the maps (as we do in Figure 3—figure supplement 2). This combination of results allows the reader to determine whether apparent differences in the independent maps are likely to be “true” differences vs differences related to an arbitrary threshold.

The reviewer is correct in noting that we directly contrasted the recollection effects for the immediate and delayed lists, which we have now stated more clearly in the Methods and Results sections. There were no clusters that survived this contrast at our voxel-wise threshold. Because we were also interested in whether this result was compatible with our ROI findings, we relaxed the threshold to investigate what was going on at the voxel-level within the posterior hippocampus. Here we observed that a large fraction of voxels in the posterior hippocampus were numerically more active during immediate than during delayed recollection, leading to an overall difference across the ROI, but no individual clusters that survived voxel-wise correction. It is worth noting that, for our purposes, the anatomical ROI approach was preferable to voxel-wise analyses because it better accounts for inter-subject anatomical variability (Nieto-Castanon et al., 2003, NeuroImage).

*2) Context similarity in anterior MTL (especially Ant HF) is a nice finding that carries a lot of weight. Ant HF represented context (associated room) for the immediate condition, and while it did not represent context for the delayed condition across the group, it appears to do so in subjects who did remember the associated room in the associative text. My main concern is that the finding is somewhat weak (authors used p < 0.05, one-tailed test, uncorrected, while making at least 8 tests (8 ROIs across left and right MTL))*.

We agree that when considered alone, the context similarity effect in the anterior HF is a small (albeit theoretically interesting) effect. Yet, it is internally consistent with the finding that activity in the anterior HF predicted recollection across both delays. Even more compellingly, it correlates with an independent behavioral measure designed to tap into the same process. It is not clear how large of an effect one would expect to see for a subtle context manipulation: recent work from our lab has shown that hippocampal pattern similarity is greatest for trials that share both object and context information (Hsieh et al., 2014; Libby et al., 2014), compared to context information alone. Future work should hone in on this result to determine its sensitivity and extensibility to other kinds of context manipulations. We have revised the manuscript to explicitly acknowledge these caveats in the Discussion section.

*3) The authors also argue that their results pose some challenge to both standard consolidation model and memory transformation model because they would not predict the differences between anterior and posterior HF. In my view, the lack of subregional specificity in those models does not necessarily mean a challenge to them. Furthermore, most consolidation literature concerns longer delays than just one day (sometimes years). Therefore, the current experiment may be tapping into delay-related changes in recollection but it is not clear if they are related to consolidation. For instance, the authors note that Post HF may code for item-specific context (not measurable here). Post HF may no longer show recollection effects after a delay because this specific context was forgotten over the delay. All what is left may be more general context (such as the associated room) represented in Ant HIP, observable at least in the subjects who remembered the associated room*.

We have revised the manuscript to clarify that the standard consolidation and memory transformation models are agnostic with respect to anterior-posterior HF differences. As such, these results do not directly “challenge” these models, but instead they suggest the need for models that account for the anterior-posterior difference.

The reviewer also raises an important point about the relationship between the current findings, which were based on a one-day delay, and changes that may unfold over longer intervals. In our revision, we have directly addressed this issue in the Discussion section:

“An important question for future research is how changes in neural representation over a one-day interval relate to the changes over longer intervals that are the focus of memory consolidation and transformation accounts. There is considerable evidence suggesting that early stages of systems consolidation can be initiated during the first night of sleep after learning (Born and Wilhelm, 2012), but little is known about how initial changes in hippocampal representation are related to longer-lasting changes. It is possible that the changes in activation magnitude and pattern similarity observed here reflect forgetting of certain aspects of context information, and that qualitatively different kinds of changes—particularly in cortical representation—might be apparent over a longer interval.”

Reviewer 3:

*It would be helpful to have figures depicting the similarity results at immediate and delay timepoints for the PHC and post HF (which are reported to not show context similarity effects and/or effects that are sustained over time) alongside those presented for ant HF and L PRC as in*
Figure 4*. It is one thing to note that the similarity effects are not present or stable, but the baseline level of similarity in PHC and post HF may be higher than in other regions, which may affect the conclusions regarding function*.

We have modified Figure 4—figure supplement 1 to show the same-room and different-room similarity estimates for all MTL ROIs. Differences in baseline similarity between regions are difficult to interpret, because they might arise from regional differences in signal intensity or functional homogeneity. This problem can be compared to the problem of interpreting between-region differences in univariate BOLD signal intensity for a single experimental condition. As typically done for univariate analyses, we resolved this problem by contrasting pattern similarity estimates between two conditions of interest (i.e., the difference between same-room and different-room similarity).

*The conclusion that similarity in Ant HF is maintained over a delay does not receive strong support in the data considering that there is no difference between same, and different, room similarity in delayed recognition. This would then have consequences for the interpretation. The basis for this conclusion of stability rests on the correlation between similarity and recognition performance, but while this addresses the contributors to performance, it does not address stability per se*.

We have revised the manuscript to clarify that anterior HF activation magnitude is relatively stable over time. We have been careful to note that, in the anterior HF, voxel pattern information related to study context remains only for those participants who retain context associations, as indicated by associative recognition performance (please see the Discussion section).

*It would also be helpful to see whether the correlation between immediate similarity and recognition is also significant*.

For the anterior hippocampal region shown in Figure 4, the correlation between context similarity and associative recognition is not significant for the immediate list, r(16)=-.17, p=.49. We now include this datapoint in the Discussion, but hesitate to interpret this null result, because there is little variability in the context similarity estimates for the immediate list except for two extreme values (see Figure 5).Author response image 1.